# Detection and Characterization of Estrogen Receptor α Expression of Circulating Tumor Cells as a Prognostic Marker

**DOI:** 10.3390/cancers14112621

**Published:** 2022-05-25

**Authors:** Retno Ningsi, Maha Elazezy, Luisa Stegat, Elena Laakmann, Sven Peine, Sabine Riethdorf, Volkmar Müller, Klaus Pantel, Simon A. Joosse

**Affiliations:** 1Department of Tumor Biology, University Medical Center Hamburg-Eppendorf, 20246 Hamburg, Germany; retno.ningsi@mail.de (R.N.); m.elazezy@uke.de (M.E.); luisa@leifheit-stegat.de (L.S.); s.riethdorf@uke.de (S.R.); pantel@uke.de (K.P.); 2Department of Gynecology, University Medical Center Hamburg-Eppendorf, 20246 Hamburg, Germany; e.laakmann@uke.de (E.L.); v.mueller@uke.de (V.M.); 3Department of Transfusion Medicine, University Medical Center Hamburg-Eppendorf, 20246 Hamburg, Germany; s.peine@uke.de; 4Mildred Scheel Cancer Career Center HaTriCS4, University Medical Center Hamburg-Eppendorf, 20246 Hamburg, Germany

**Keywords:** circulating tumor cells (CTCs), estrogen receptor (ER)

## Abstract

**Simple Summary:**

CTCs are considered the seeds of metastases and their presence in the blood is related to survival in breast cancer. Estrogen receptor (ER) is considered a direct target of hormonal therapy. Evaluating the expression of ER in CTCs to characterize the tumor and assess therapy efficacy might lead to individualized treatment. In ER-positive metastatic breast cancer patients, CTCs were detected in half of the patients. Most of these patients carry ER-negative CTCs only, whereas a third shows a mixture of ER-positive and -negative CTCs. Shorter relapse-free survival was associated with CTC-positivity. The CTC-ER expression was intra- and inter-patient heterogeneous, highlighting potential endocrine therapy resistance. Therefore, monitoring ER-CTC status in advanced breast cancer could add a prognostic value to CTC enumeration and may serve as a predictive marker for therapeutic resistance, which may need to be addressed on a larger scale in future studies.

**Abstract:**

CTCs have increasingly been used as a liquid biopsy analyte to obtain real-time information on the tumor through minimally invasive blood analyses. CTCs allow for the identification of proteins relevant for targeted therapies. Here, we evaluated the expression of estrogen receptors (ER) in CTCs of patients with metastatic breast cancer. From sixty metastatic breast cancer patients who had ER-positive primary tumors (range of 1–70% immunostaining) at initial cancer diagnosis, 109 longitudinal blood samples were prospectively collected and analyzed using the CellSearch System in combination with the ERα monoclonal murine ER-119.3 antibody. Prolonged cell permeabilization was found to be required for proper staining of nuclear ER in vitro. Thirty-one cases were found to be CTC-positive; an increased number of CTCs during endocrine and chemotherapy was correlated with disease progression, whereas a decrease or stable amount of CTC number (<5) during treatment was correlated with a better clinical outcome. Survival analyses further indicate a positive association of CTC-status with progression-free survival (HR, 66.17; 95%CI, 3.66–195.96; *p* = 0.0045) and overall survival (HR, 6.21; 95%CI, 2.66–14.47; *p* < 0.0001). Only one-third of CTC-positive breast cancer patients, who were initially diagnosed with ER-positive primary tumors, harbored ER-positive CTCs at the time of metastasis, and even in those patients, both ER-positive and ER-negative CTCs were found. CTC-positivity was correlated with a shorter relapse-free survival. Remarkably, ER-negative CTCs were frequent despite initial ER-positive status of the primary tumor, suggesting a switch of ER phenotype or selection of minor ER-negative clones as a potential mechanism of escape from ER-targeting therapy.

## 1. Introduction

Breast cancer is the most common cancer among women worldwide, with over 2.3 million new cases in 2020 and more than 685,000 deaths in the same year [1]. One of the most essential biomarkers in breast cancer management is the estrogen receptor (ER). More than 70% of breast cancer cases express ER, and the classification of breast cancer into ER-positive and ER-negative determines the type of therapy the patient will receive. Estrogen is a steroid hormone that upon binding to its receptor in the normal situation, affects the growth and differentiation of the mammary gland. In breast cancer, estrogen stimulates tumor cell proliferation.

In general, patients with ER-positive breast cancer respond well to endocrine therapy due to suppressing estrogen production or blocking the receptor’s binding site [2]. In addition, as ER-positive breast cancer has shown to exhibit hyperactivity of cyclin-dependent kinase 4 and 6 (CDK-4/6), a combination of endocrine therapy with CDK-4/6 inhibitors has been shown to further improve outcome [3]. Unfortunately, 10% of breast cancer cases are diagnosed with metastatic disease and many patients will develop systemic relapse over time. Although the hormone receptor status of relapses are usually maintained due to crosstalk between hormone receptors and growth factors, but also due to genetic progression and point mutations [4], many patients develop therapy resistance after 24–36 months [5]. As a consequence, distant metastasis is still the leading cause of breast cancer-related death [6].

Cancer metastasis starts with single or clusters of tumor cells separating from the primary tumor and intravasating into the bloodstream [7]. The motility of these cells enabling tumor dissemination is made possible by the epithelial-to-mesenchymal transition [8]. Next, these so-called circulating tumor cells (CTCs) may extravasate into distant organs, adapt to a new environment, and grow out to become a new tumor mass. In breast cancer, as well as many other cancer entities, it has been shown that the number of detectable CTCs in the blood of early-stage and metastatic breast cancer patients is negatively associated with progression-free survival [9]. Because blood is easily acquired and can be obtained repeatedly, CTCs as a so called “liquid biopsy” have increasingly gained attention during the past decades [7]. Phenotyping CTCs can provide crucial information on the evolving characteristics of the tumor during the progression and development of therapy resistance [10,11,12]. Although CTCs can be detected in more than 60% of metastatic breast cancer patients [13,14], due to the low amount of CTCs (typically 1–10 CTCs/7.5 mL blood), reliable detection can still be challenging [7,15,16]. To overcome this challenge, many techniques for the quantification, characterization, and isolation of CTCs have been developed based on different cell properties [7,17,18,19,20]. Currently, the only FDA-cleared method is the CellSearch^®^ System, which is therefore considered the gold standard in CTC detection [10,21]. This system enriches EpCAM-positive cells and detects CTCs based on the expression of keratin (K), but negatively selects for CD45 expression. Besides the initial number of CTCs, the change in the status of CTCs during treatment with systemic therapy in metastatic breast cancer patients has recently been linked with prognosis in a meta-analysis by Yan et al. [22], showing the relevance of monitoring treatment efficacy.

Previously, we have found that ER expression among CTCs is heterogeneous in metastatic breast cancer patients who were initially diagnosed with an ER-positive primary breast tumor [23]. The heterogeneous expression of ER may indicate estrogen independence of disseminated cells and thus a possible predictor for hormonal treatment failure. Not surprisingly, in a more recent study, it was shown that a shift from ER-positve breast cancer to ER-negative CTCs in metastatic disease was associated with a worse prognosis as compared to cases in which ER-positivity was stable [24]. Paoletti and colleagues developed a CTC-Endocrine Therapy Index (CTC-ETI) using the semi-automated CellSearch System for CTC quantification and characterization by assessing the expression of ER, BCL-2, ErbB2, and Ki-67 [25,26,27]. The preliminary data from Paoletti et al., demonstrated the reproducibility of CTC-ETI as a predictive factor for resistance to endocrine therapy in ER-positive metastatic breast cancer patients [25]. Recently, they determined the CTC-ETI in a larger cohort of ER-positive metastatic breast cancer patients in their phase 2 trial [27]. Paoletti and colleagues found that patients with high CTC-ETI were more likely to have rapid progression after three months of treatment. Taken together, it may be concluded that ER expression on CTCs have a predictive value of endocrine therapy response.

In the presented study, we evaluated the ER-CTC status using the ERα monoclonal murine ER-119.3 antibody, which to our knowledge, has been tested only once so far. We aimed to reproduce the previously published staining conditions in vitro, followed by testing a cohort of metastatic breast cancer patients with ER-positive primary tumor using the CellSearch System for the quantification of CTCs in metastatic breast cancer. Based on the available literature, we expected an ER-positivity rate of approximately 30% among CTCs. Determination of the ER status of CTCs could help in therapy stratification, monitoring therapy efficacy, and predicting the risk for metastatic relapse.

## 2. Materials and Methods

### 2.1. Cell Lines

The two human breast cancer cell lines MCF-7 and SK-BR-3 (acquired from ATCC), were used in this study as positive and negative control for estrogen receptor (ER) expression to optimize the experimental conditions. Both cell lines were cultivated in DMEM (catalog no. E15-011, PAA Laboratories, BioPath Stores, Cambridge, UK) at 37 °C and 10% CO_2_. The medium was supplemented with 10% fetal bovine serum, penicillin-streptomycin antibiotics, and L-Glutamine (catalog no. E15-151, M11-004, P11-010, PAA Laboratories).

### 2.2. Patients

Sixty metastatic breast cancer patients with initially ER-positive primary tumors were included in the study from July 2015 to December 2020. Patients were treated for metastatic breast cancer at the University Medical Center Hamburg-Eppendorf and received endocrine therapy as first-line therapy, followed by chemotherapy at the progression of the disease. The mean age of patients was 62 years (range: 34–86). All patients gave written informed consent to be included in the study. This study was approved by the local ethical board (Ärztekammer Hamburg) under number PV4367.

### 2.3. CTC Detection

The CellSearch CXC kit (Menarini Silicon Biosystems, Bologna, Italy) was used to detect circulating tumor cells. This kit contains ferrofluid particles coated with anti-EpCAM antibodies, anti-keratin antibodies recognizing keratins (fluorescein-labeled) to specifically identify epithelial cells, an antibody against CD45 (labeled with allophycocyanin) as a negative selection marker for white blood cells, and a nuclear dye 4’,6-diamidino-2-phenylindole (DAPI) [28]. We added ERα monoclonal murine ER-119.3 antibody (phycoerythrin-labeled, Ex_max_ 496 nm/Em_max_ 578 nm) from (Janssen R&D) to identify estrogen receptor expression in the nucleus.

The enrichment was done automatically by the CellTracks Autoprepsystem. The enriched cells were collected in the MagNest cartridge. Using CellSpotter Analyzer, each cell in the cartridge was displayed based on the signal sent from the antibodies. Cells were enumerated as circulating tumor cells if the signal for keratin and cell nucleus were positive and the signal for CD45 was negative. The estrogen receptor expression was shown by the positive nuclear signal of ERα antibody.

### 2.4. Statistics

The statistical analysis was performed with R (version 4.0.1) [29] and In-Silico Online, version 2.3.0 [30]. The difference in means was assessed by Welch’s two-sided *t*-test, the difference in the median by Wilcoxon’s signed-rank test with continuity correction, whereas the association between categorical variables was tested using the multinomial exact test [31]. Survival analyses were performed using the log-rank test and multivariable analysis by Cox proportional hazards function, with death by cancer as an endpoint. An alpha level of 0.05 was applied to determine statistical significance. Arithmetic mean values are presented together with standard deviations (*s*).

## 3. Results

### 3.1. In Vitro Evaluation of the ER Antibody

In order to evaluate previously published results on the ER antibody for semi-automated characterization of CTCs [10], an ER-positive (MCF-7) and ER-negative cell line (SK-BR-3) were cultured under identical conditions and used to establish the optimal experimental settings. Eight blood samples from healthy donors were spiked with 100 tumor cell line cells cultured at 37 °C and 10% CO_2_ with no incubation time after spiking; four blood samples were spiked with MCF-7 cells and four with SK-BR-3 cells. Using the CellSearch System; the mean recovery of the cells was 94% (*s* = 7%) and 89% (*s* = 6.1%) of the MCF-7 and SK-BR-3 cells, respectively. However, only 15% (*s* = 2.1%) of the MCF-7 cells were found to be ER-positive on average. As expected, none of the detected SK-BR-3 cells were ER-positive.

Because more ER-positive MCF-7 cells were expected, the experiment was repeated, but spiked blood was incubated in CellSave tubes containing preservative buffer for approximately 24 h to fix and permeabilize the cells. The mean percentage of ER-positive tumor cells among the total detected MCF-7 cells increased to 45% (*s* = 2.2), and among SK-BR-3 cells stayed at 0% (Figure 1). In addition, we validated the results by determining ER protein expression using our previously published protocol as well [9]. The mean percentage of ER-positive MCF-7 cells as detected with our manual protocol was 40% (*s* = 4.3), which was comparable with the results obtained with the CellSearch System (mean: 45%, *s* = 5.1; *p*-value: 0.19, Welch’s two Sample *t*-test).

Based on these results, the final protocol was as follows: blood from metastatic breast cancer was drawn into a CellSave preservative tube and incubated at room temperature until processing the next day. The ER channel was analyzed using 0.2-s integration time; higher integration time resulted in too high a background.

### 3.2. Circulating Tumor Cells (CTCs) in Metastatic Breast Cancer Patients

Patients included in this study were selected prospectively based on the diagnosis of advanced disease with distant metastasis. Primary diagnosis of breast cancer among these patients was at a mean age of 52 years (range: 28–86), on average 7 years (range: 0–33) before study inclusion (Table 1). All patients were diagnosed with ER-positive primary breast cancer and twelve (20%) with primary metastasis. One hundred and nine blood samples from 60 patients were collected during this study. Nine samples were collected at the admission interview before the initiation of systemic therapy, 37 samples were collected during endocrine therapy, 51 samples were collected at a progressive stage at which the patients received chemotherapy, and 12 samples failed the analysis. Out of all 97 blood samples, CTCs were found in 31; these blood samples were from 20 patients (Table 1 and Appendix A). In 15% (15/97) of the samples, 1–4 CTCs/7.5 mL were found, and in 16% (16/97) of samples, 5 or more CTCs could be detected. The presence of CTCs was not correlated to the TNM-stage at primary diagnosis or therapy at blood draw but was correlated to the stage of disease at the time of blood draw. Patients experiencing progression of disease were more frequently diagnosed with CTCs in their blood compared to patients with stable disease (*p* = 0.0476, multinomial exact test).

### 3.3. Evaluation of ERα Monoclonal Murine ER-119.3 Antibody on CTCs

ER-positive CTCs could be detected in 10 out of 31 CTC-positive blood samples (32%). In these ten cases, the number of detected CTCs ranged from 1 to 207 of which the mean percentage of ER-positive CTCs was 28% (range: 9–100%; Figure 2). Of all detected CTCs in this study (*n* = 1485), 18% (*n* = 268) were ER-positive. CTCs were detected in 4/9 (44%) of the samples that were collected before therapy; of these 2/4 (50%) cases were diagnosed with ER-positive CTCs. Among 37 samples obtained during hormonal therapy, in 27%, (10/37) CTCs could be detected, of which 1/10 (10%) was ER-positive. In 17/52 (33%) of the blood samples from patients treated with chemotherapy, CTCs were detected, and 7/17 contained ER-positive CTCs. Although the fewest ER-positive CTCs could be detected among the patients treated with hormone therapy, no statistical significance was found (*p* = 0.0985, multinomial exact test). These results indicate a heterogeneous expression of ER among CTCs within individual patients.

### 3.4. Monitoring CTC Count during Therapy

Longitudinal blood samples were obtained from twenty-five patients. Three patients were initially diagnosed with ER-positive CTCs but changed to completely CTC-negative or ER-negative CTCs during the course of the study (Appendix A). All other patients were diagnosed with ER-negative CTCs only according to the CellSearch System results. If more than two blood draws were taken, only the first two before and after a change of clinical response were considered for further analyses.

Fourteen patients experienced progression of the disease during this study. At the time of blood sampling before progression, two patients were found to be positive for CTCs. During the second blood sampling after the diagnosis of progression, one CTC-positive case was diagnosed CTC-negative, and one stayed positive (Appendix A); five CTC-negative cases converted to CTC-positive, and seven remained CTC-negative. The increase in the median number of CTCs was significantly different (*p*-value: 0.0367, paired Wilcoxon signed-rank test with continuity correction), as was the number of cases converting CTC-status from the first to second blood draw (*p*-value: 0.0352, exact McNemar test).

Eleven cases were diagnosed with the stable disease throughout the period of the study or converted from progression to stable disease. At the first blood collection, seven patients were diagnosed CTC-positive and 4 CTC-negative. At the second blood draw, one case remained positive, whereas the other ten cases became or stayed CTC-negative. The decrease in the number of CTC from the first to the second blood draw was statistically significant (*p*-value: 0.0111, paired Wilcoxon signed-rank test with continuity correction), as was the number of cases converting to a CTC-negative status (*p*-value: 0.0156, exact McNemar test).

Because these data suggest that an increase or decrease of the median number of CTCs is associated with progression and stable disease, respectively, the median differences between the two blood draws of the two clinical response groups were compared (Figure 3). The median difference in CTC number between blood draws of the patients in stable disease was significantly less compared to the median difference in CTC number between blood draws of the patients in the progression of disease (*p*-value: 0.0021, Wilcoxon rank-sum test with continuity correction). Among the patients with stable disease, a decrease in CTCs was seen in six and no change in five patients. Among the patients experiencing a progression of disease, a decrease in CTC number was seen in 1 patient, an increase in 6 patients, and no change in 7 patients (*p*-value: 0.0339, multinomial exact test). In order to further study the clinical association with change in CTC status, survival analyses were performed next.

### 3.5. Survival Analysis

The median follow-up after the first blood draw was 22.4 months (range: 1.8–61.6); during the time of the study, 28 patients died (47%). The patient data were divided on CTC status (positive or negative) at the time of the first blood draw, and overall survival was compared. More than 50% of the patients characterized as initially CTC-negative had an overall survival longer than the duration of the study, whereas the median survival of the patients diagnosed as CTC-positive was 8.6 months (*p*-value < 0.0001, log-rank test, Figure 4A). Additionally, in uni- and multivariable analyses, CTC-status was independently associated with survival (HR: 66.2, 95%CI: [3.7, 1196], *p*-value: 0.0045), whereas T-, N-, and M-stage at primary diagnosis were not (Table 2). These data indicate that a positive diagnosis for CTCs is highly associated with poor overall survival.

The 25 patients from which multiple blood samples were collected were separated into two groups: (1) cases with a stable CTC-status or conversion from positive to negative CTC-status and (2) cases with the conversion from negative to positive CTC-status. The individuals experiencing a CTC conversion (*n* = 6) had a median survival of 13.5 months, whereas the individuals with a stable CTC status (*n* = 19) had a median survival of 49.3 months, (*p*-value: 0.053, log-rank test, Figure 4B). In only three of the cases in which multiple samples were collected, ER+ CTCs were detected. Therefore, the effect of CTC ER-status on survival could not be assessed longitudinally. When considering only the first blood samples, the median overall survival of patients diagnosed with ER-positive CTCs was 7.3 months and 12.5 months of patients diagnosed with ER-negative CTCs only (*p*-value: 0.32, log-rank test).

## 4. Discussion

CTCs are considered the seeds for new metastases and their presence in the blood of patients with primary breast cancer may provide prognostic information in regards to relapse-free survival. Similarly, in a metastatic setting, the quantification of CTCs may offer prognostic information in regards to overall survival. Moreover, determining the expression of therapeutic markers to characterize the tumor as well as to monitor therapy efficiency could lead to personalized treatment regimens.

Earlier studies have shown that progression-free and overall survival of metastatic breast cancer are correlated with CTC presence (PFS: HR, 1.78; 95% CI, 1.52–2.09; OS: HR, 2.33; 95% CI, 2.09–2.60), irrespective of the CTC detection method and time point of blood withdrawal [32]. In line with these data, our study shows a clear correlation with overall survival and the detection of CTCs using the CellSearch System as well. Nevertheless, it should be noted that the sample cohort of our study is heterogeneous and the blood was collected at various time points in the patients’ therapy regiments, and confounding factors cannot be excluded. Therefore, an important feature of liquid biopsy that can be taken advantage of is repeated measurement and monitoring a patient’s CTC status during the course of therapy. In our study, fourteen patients experienced progression, whereas ten patients remained stable or even had a regression. As expected, patients with progression showed conversion of being CTC negative to positive or were diagnosed with an increase in the number of CTCs. Once the therapy was escalated from endocrine therapy to chemotherapy, we observed a decrease in CTC number. Taken together, our data are in line with earlier findings in which a significant correlation was found between CTCs and progression-free survival [22,33].

The majority of clinical studies on breast cancer involving liquid biopsy are based on CTC quantification only. Although a positive diagnosis for CTCs is strongly associated with poor overall survival in metastatic breast cancer patients, CTC enumeration alone cannot elucidate the molecular mechanisms involved in therapy resistance or guide the choice of systemic therapy. Therefore, phenotyping CTCs can add another dimension to cancer management that may ultimately lead to personalized treatment. Especially the estrogen receptor (ER) expression status is an important therapeutic marker in the management of breast cancer. Forsare et al., showed that the presence of ER-positive CTCs at baseline and after initiation of systemic treatment is associated with a better prognosis as compared to ER-negative CTCs in patients with breast cancer [24]. The authors use a manual ER staining after CellSearch System-based enrichment of CTCs and apply methanol for fixation and permeabilization, which is in contrast to the study of Paoletti et al. [27] and the study presented here, where ER labeling is performed within the CellSearch System. Nevertheless, in all studies, approximately in only a third of the CTC-positive cases can ER-positive CTCs be detected: 36% (38/107), 25% (16/63), and 32% (10/31), respectively [24,27]. In all studies, heterogeneity in ER expression status among CTCs within individual patients can be observed, consistent with our previous study in which we did not use the CellSearch System for CTC enrichment [23].

As a consequence of the limitations of the study, the relatively low number of patients with detectable ER-positive CTCs, the relatively short follow-up time, and heterogeneity in sample collection time-points in our study, an association with survival or clinical variables such as tumor burden and grade could not be made with ER status of CTCs yet. Furthermore, downregulation of EpCAM in CTCs that underwent epithelial–mesenchymal transition (EMT) may result in missing critical tumor cell subpopulations. Previously, we established an alternative approach based on EpCAM-free detection to overcome this limitation and capture as many CTCs as possible [23]. Nevertheless, to our knowledge, this is one of few studies that has evaluated ER status of CTCs using the CellSearch CXC kit. An important finding of this study is the requirement of permeabilization in order for ER antibodies to penetrate the cell membrane and reach the cell nucleus. Therefore, future studies conducted using the CellSearch System should make use of the CellSave preservation tubes for blood collection. Furthermore, (pre-)clinical studies may use the results of this study for sample-size calculations, taking the expected fraction of patients with detectable ER-positive CTCs of 32% into account. Recently, we established a permanent CTC line (CTC-ITB-01) derived from ER-positive breast cancer patients. Performing this cell line will facilitate the study of ER expression on a large scale providing further insights into the dynamics of ER expression in therapy response [34]. Overall, the identification and monitoring of ER status of CTCs is extremely important for the management of breast cancer patients. Further investigation towards a robust and reproducible assay is needed with larger cohorts.

## 5. Conclusions

Half of metastatic breast cancer patients that were diagnosed with an ER-positive primary tumor harbor detectable CTCs in their peripheral blood circulation. Most of these patients carry ER-negative CTCs only, whereas approximately a third show a mixture of ER-positive and -negative CTCs. The detection of CTCs remains a well-established marker for poor outcomes and shorter overall survival. Monitoring ER-CTC status could add a prognostic value to CTCs enumeration and may serve as a therapy resistance prediction. Therefore, further studies with large cohorts on long-term monitoring of ER-CTCs status are required to address the predictive value of ER-CTC for endocrine therapy efficacy in breast cancer patients.

## Figures and Tables

**Figure 1 cancers-14-02621-f001:**
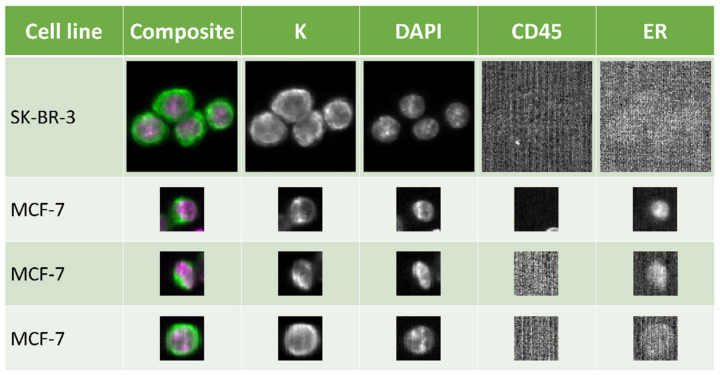
ER expression detected in breast cancer cells by the CellSearch System. SK-BR-3 (top row) is an ER-negative cell line and was used as a negative control. MCF-7 is ER-positive and examples are given for strongly positive, (second row), weakly positive, (third row), and (bottom row) nuclear staining. Standard selection markers of the CellSearch System for tumor cells are keratin (K) and DAPI, whereas CD45 is used as an exclusion marker. Automatic low signal compensation leads to high background signal seen in CD45 and ER-negative scans. Pictures were taken using 10× magnification.

**Figure 2 cancers-14-02621-f002:**
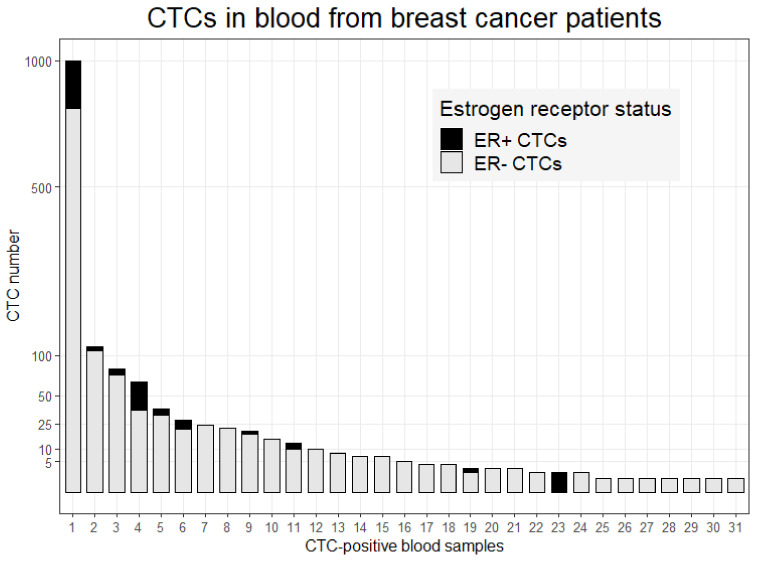
Number of ER-positive and -negative CTCs. Patient blood samples in which CTCs were detected (*n* = 31). Depicted are the number of ER-positive (black) and ER-negative (gray) CTCs per sample.

**Figure 3 cancers-14-02621-f003:**
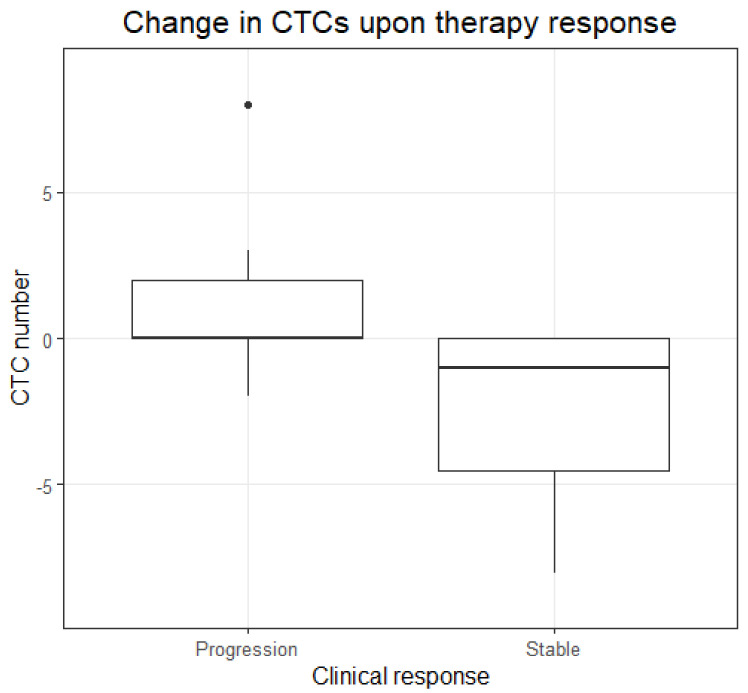
Change in CTC number. Boxplots showing the distribution of the change in the number of CTCs at progression of the disease (*n* = 14) and at stable disease (*n* = 11).

**Figure 4 cancers-14-02621-f004:**
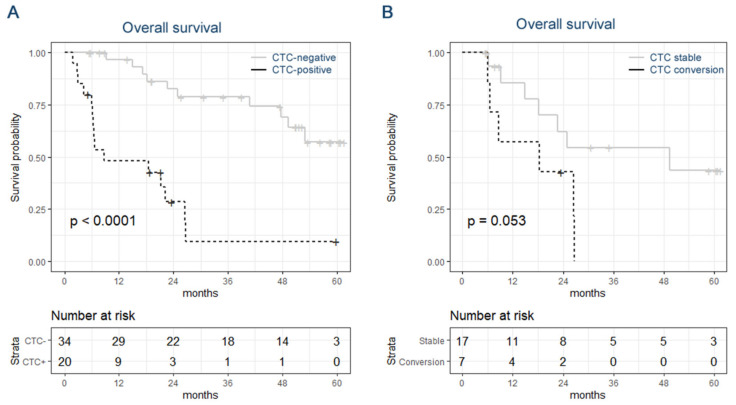
Kaplan–Meier curves. Overall survival of breast cancer patients based on CTC-status (**A**) and based on stable CTC-status or CTC conversion (**B**).

**Table 1 cancers-14-02621-t001:** Demographic statistics. Number of CTC-positive (CTC+) and CTC-negative (CTC-) blood samples divided according to clinical variables of the patients at primary diagnosis. *p*-values were calculated using Welch’s two-sided *t*-test and the multinomial exact test. CTC values were not available for three patients.

At Primary Diagnosis	N = 60	CTC+ (*n* = 20)	CTC− (*n* = 37)	*p* Value
**Age (years)**				
Mean	52	55	49	0.0848
Range	28–76	39–76	28–75	
**ER**				
Positive	60	20	37	-
**PR**				
Positive	54	18	34	0.9411
Negative	6	2	3	
**ERBB2**				
Positive	8	2	5	0.8235
Negative	43	15	26	
**Grade**				
G1–2	29	11	18	0.4656
G3	10	2	7	
**T-stage**				
1–2	34	12	21	0.7028
3–4	12	3	8	
**N-stage**				
0	12	4	7	0.9613
1–3	33	11	21	
**initial M-stage**				
0	13	3	10	0.8683
1	12	2	9	
**At blood draw**	**N = 97**	**CTC+ (*n* = 31)**	**CTC− (*n* = 66)**	
**Age (years)**				
Mean	62	63	62	0.8006
Range	34–86	39–86	34–80	
**Therapy**				
Naïve	9	4	5	0.5536
Endocrine	37	10	27	
Chemo	51	17	34	
**Stage**				
Stable	37	7	30	0.0476
Progression	38	17	21	

**Table 2 cancers-14-02621-t002:** Cox proportional hazard ratios. Estimated coefficients of overall survival on breast cancer subjects. Calculated are the corresponding hazard ratio (HR), 95% confidence interval (CI) of the hazard ratio, and *p*-value in uni- and multivariable Cox proportional hazard analysis for CTC-status with negative as reverence, and T-, N-, and M-stage at initial diagnosis.

	Univariable Analysis				Multivariable Analysis			
Covariate	Coefficient (b_i_)	HR [exp(b_i_)]	HR 95% CI	*p*-Value	Coefficient (b_i_)	HR [exp(b_i_)]	HR 95% CI	*p*-Value
CTC-positive	1.83	6.21	2.66, 14.47	0.0002	4.19	66.17	3.66, 1195.96	0.0045
T3–4 (rev: T1–2)	−0.17	0.84	0.31, 2.29	0.73	0.47	1.59	0.12, 20.54	0.72
N1 (rev: N0)	−0.30	0.74	0.27, 2.07	0.57	0.26	1.30	0.22, 7.73	0.77
M1 (rev: M0)	−0.71	0.49	0.15, 1.61	0.24	−1.92	0.15	0.01, 1.50	0.11

## Data Availability

The datasets generated during and/or analyzed during the current study are available from the corresponding author on request.

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
