# Peer review of "Detection and Characterization of Estrogen Receptor α Expression of Circulating Tumor Cells as a Prognostic Marker"

_cancers, 2022, doi:10.3390/cancers14112621_

Round 1

Reviewer 1 Report

keep up the good work

Author Response

Point-by-point reply to the comments raised by the reviewers with respect to the manuscript " Detection and characterization of estrogen receptor α expression of circulating tumor cells as a prognostic marker" by Ningsi et al.

We would like to thank the reviewers for their time and effort to provide their reviews. We have carefully revised the manuscript according to the suggestions provided.

Reviewer 1 - Comments for the Author.

keep up the good work

We thank the reviewer for the motivating comment.

Reviewer 2 Report

This article focuses on evaluating the expression of therapeutic marker to characterize tumors using CellSearch platform to detect CTCs with high sensitivity, specificity and reproducibility in small volumes of blood. The authors make several interesting observations which have immediate translational applications but more clarification of the data is required which will give ER positive CTC, a prognostic marker status.

  1. The abstract can be edited to include more information about presence of ER positive CTC and how it is associated with increased tumor burden and grade. Was an optimal prognostic threshold applied with respect to progression-free survival? If so, it needs to be clearly mentioned.
  2. Figure 1 needs more clarification as to why there are 3 panels of images for MCF-7 as compared to SK-BR-3 and also how these images are analyzed to deduce the conclusion as mentioned in lines 178-180.
  3. It would be beneficial to know the range in years (eg. Aug 2015-Jan 2020) that the patients were recruited in.
  4. In line 216, how did the authors get 37 samples and 10/37 is not 33%. This entire paragraph needs thorough reviewing and rewriting with the inclusion of Figure 2, which is missing so its more comprehensible.
  5. Table 2 has been mentioned as table 1.

Author Response

Point-by-point reply to the comments raised by the reviewers with respect to the manuscript " Detection and characterization of estrogen receptor α expression of circulating tumor cells as a prognostic marker" by Ningsi et al.

We would like to thank the reviewers for their time and effort to provide their reviews. We have carefully revised the manuscript according to the suggestions provided.

All modifications in the revised manuscript are marked up using the "Track Changes" function.

Reviewer 2 - Comments for the Author.

This article focuses on evaluating the expression of therapeutic marker to characterize tumors using CellSearch platform to detect CTCs with high sensitivity, specificity and reproducibility in small volumes of blood. The authors make several interesting observations which have immediate translational applications but more clarification of the data is required which will give ER positive CTC, a prognostic marker status.

  1. The abstract can be edited to include more information about presence of ER positive CTC and how it is associated with increased tumor burden and grade. Was an optimal prognostic threshold applied with respect to progression-free survival? If so, it needs to be clearly mentioned.

In response to the reviewer's comment, the presence of ER-positive CTC was elucidated in the abstract on page 1 as follows:

"Only one-third of CTC-positive breast cancer patients, who were initially diagnosed with ER-positive primary tumors, harbored ER-positive CTCs at the time of metastasis, and even in those patients, both ER-positive and ER-negative CTCs were found."

The ER-positive CTCs and their association with tumor burden and grade was not conclusive enough due to the low number of patients with detectable ER-positive CTCs. In response to the reviewer's comment, we have therefore addressed this point in the Discussion on page 10 as follows:

"As a consequence of the limitations of the study, the relatively low number of patients with detectable ER-positive CTCs, the relatively short follow-up time, and heterogeneity in sample collection time-points in our study, an association with survival or clinical variables such as tumor burden and grade could not be made with ER status of CTCs yet."

With respect to an optimal prognostic threshold, the range of CTC counts was too narrow to perform such an analysis and a cut-off of 1 CTC was used as has been shown to be clinically relevant.  The detection of a single CTC in 1 tube of blood from breast cancer patients by the CellSearch System has been shown to have a strong prognostic value [Janni et al, Clin Canc Res, 2016]. In this pooled study, 3173 breast cancer patients were included from five different breast cancer institutions and different cut-offs from 1 until 20 CTCs were used. A single CTC is shown to significantly increase the hazards ratio in overall and disease-free survival.

Similarly, our study presented here, shows a strong correlation between overall survival and the presence of one or more CTCs.

  1. Figure 1 needs more clarification as to why there are 3 panels of images for MCF-7 as compared to SK-BR-3 and also how these images are analyzed to deduce the conclusion as mentioned in lines 178-180.

In order to respond to the reviewer's comment, we adjusted the legend of Figure 1 as follows:

"Figure 1 - ER expression detected in breast cancer cells by the CellSearch System. SK-BR-3 (top row) is an ER-negative cell line and was used as a negative control. MCF-7 is ER-positive and examples are given for strongly positive (second row), weakly positive (third row), and negative (bottom row) nuclear staining. Standard selection markers of the CellSearch System for tumor cells are keratin (K) and DAPI, whereas CD45 is used as an exclusion marker. Automatic low signal compensation leads to high background signal seen in CD45 and ER-negative scans. Pictures were taken using 10X magnification.”

We showed only one image with four cells representing the homogeneously ER-negative SK-BR-3 cells.

Furthermore, we have addressed the following point in the Result section on page 4 as follows:

"In addition, we validated the results by determining ER protein expression using our previously published protocol as well [9]. The mean percentage of ER-positive MCF-7 cells as detected with our manual protocol was 40% (s=4.3), which was comparable with the results obtained with the CellSearch System (mean: 45%, s=5.1; p-value: 0.19, Welch’s Two Sample t-test).

Based on these results, the final protocol was as follows: blood from metastatic breast cancer was drawn into a CellSave preservative tube and incubated at room temperature until processing the next day. The ER channel was analyzed using 0.2-second integration time; higher integration time resulted in too high a background."

  1. It would be beneficial to know the range in years (eg. Aug 2015-Jan 2020) that the patients were recruited in.

To respond to the reviewer's suggestion, we have added the information regarding the dates of the period in which patients were recruited in the Materials and Methods on page 3 as follows:

"Sixty metastatic breast cancer patients with initially ER-positive primary tumors were included into the study from July 2015 to Dec 2020."

  1. In line 216, how did the authors get 37 samples and 10/37 is not 33%. This entire paragraph needs thorough reviewing and rewriting with the inclusion of Figure 2, which is missing so its more comprehensible.

In response to the reviewer's comment, we have corrected the calculated percentage and adapted the paragraph in the Result section on page 6 as follows:

“ER-positive CTCs could be detected in 10 out of 31 CTC-positive blood samples (32%). In these ten cases, the number of detected CTCs ranged from 1 to 207 of which the mean percentage of ER-positive CTCs was 28% (range: 9-100%; Figure 2). Of all detected CTCs in this study (n=1485), 18% (n=268) were ER-positive. CTCs were detected in 4/9 (44%) of the samples that were collected before therapy; of these 2/4 (50%) cases were diagnosed with ER-positive CTCs. Among 37 samples obtained during hormonal therapy, in 27% (10/37) CTCs could be detected, of which 1/10 (10%) was ER-positive. In 17/52 (33%) of the blood samples from patients treated with chemotherapy, CTCs were detected, and 7/17 contained ER-positive CTCs. Although the fewest ER-positive CTCs could be detected among the patients treated with hormone therapy, no statistical significance was found (p=0.0985, multinomial exact test). These results indicate a heterogeneous expression of ER among CTCs within individual patients.”

Regarding Figure 2, We thank the reviewer for spotting this oversight, Figure 2 was added in the main text of the manuscript on page 6.

  1. Table 2 has been mentioned as table 1.

We thank the reviewer for spotting this error, we have adapted the number of Table 2 on page 8 as follows:

"Table 2 - Cox proportional hazard ratios. Estimated coefficients of overall survival on breast cancer subjects. Calculated are the corresponding hazard ratio (HR), 95% confidence interval (CI) of the hazard ratio, and p-value in uni- and multivariable Cox proportional hazard analysis for CTC-status with negative as reverence, and T-, N-, and M-stage at initial diagnosis."

Reviewer 3 Report

The article ’Detection and Characterization of Estrogen Receptor α Expression of Circulating Tumor Cells as a Prognostic Marker’ by Ningsi and Elazezy et al investigated the role of circulating breast cancer cells in 60 ER treated metastatic breast cancer patients.  While the topic is interesting, many published studies exist on the same topic (including their own previous publication); therefore, the translational novelty of the study is relatively low. On the other hand, technical advancements to current detection strategies are needed. In keeping with this, the authors aimed to validate the use of ER‐α monoclonal murine ER‐119.3 antibody for quantitative assessment of CTCs. However, there are flaws in presentation and interpretation of the presented data.

Specific comments:

  1. Inroduction: cancer statistics are outdated.
  2. One-sided test is not justified based on the one-directional expected change.
  3. The authors must take care of proper abbreviations and explanations. E.g. on line 161 what is denoted by s (s=7%, s=6.1%)?
  4. Figure 1: Why are auto-fluorescence images are shown for K, ER, and CD45 when the cells were actually labelled with fluorescent dyes? In addition, the background correction across the images seems to be different. Is this due to permeabilization? The authors may provide more technical description / validation regarding the improved detectability.
  5. Table 1: G-test is not appropriate choice for such small dataset. The test is recommended for large datasets (>1000 values) with independent samples. Please use chi-square test.
  6. Table 1: Please fix typos (e.g. 3-Jan, 4-Mar etc…)
  7. Figure legends must have more details e.g. magnification, number of values, patients, what type of sample (baseline or under therapy) etc…
  8. Figure 2. is missing!

Author Response

Point-by-point reply to the comments raised by the reviewers with respect to the manuscript " Detection and characterization of estrogen receptor α expression of circulating tumor cells as a prognostic marker" by Ningsi et al.

We would like to thank the reviewers for their time and effort to provide their reviews. We have carefully revised the manuscript according to the suggestions provided.

All modifications in the revised manuscript are marked up using the "Track Changes" function.

Reviewer 3 - Comments for the Author.

The article' Detection and Characterization of Estrogen Receptor α Expression of Circulating Tumor Cells as a Prognostic Marker' by Ningsi and Elazezy et al investigated the role of circulating breast cancer cells in 60 ER treated metastatic breast cancer patients. While the topic is interesting, many published studies exist on the same topic (including their own previous publication); therefore, the translational novelty of the study is relatively low. On the other hand, technical advancements to current detection strategies are needed. In keeping with this, the authors aimed to validate the use of ER‐α monoclonal murine ER‐119.3 antibody for quantitative assessment of CTCs. However, there are flaws in presentation and interpretation of the presented data.

  1. Inroduction: cancer statistics are outdated.

In order to respond to the reviewer's comment, we updated the breast cancer statistics in the Introduction on page 2 as follows:

"Breast cancer is the most common cancer among women worldwide with over 2.3 million new cases in 2020 and more than 685 000 deaths in the same year [1]."

  1. One-sided test is not justified based on the one-directional expected change.

The choice of an one- or two-sided test depends on the hypothesis tested. Two-sided tests are used when the null hypothesis is  vs the alternative hypothesis . In the case of our study, this would mean testing whether the median CTC number would be more OR less. However, because many studies have shown a strong positive correlation of the number of CTCs with progression of disease [e.g., Bidard, Pantel, et al. JNCI 2018], as well as the clinical and biological reasoning that an increase of tumor spread will only occur upon therapy failure, a one-sided test would be justified here. Tested is  vs the alternative hypothesis  upon therapy failure leading to disease progression. Nevertheless, we recognize the reviewer’s concern and have recalculated the statistical significance throughout the manuscript. Accordingly, the Materials and Methods, section Statistics on page 3, was adjusted as follows:

The statistical analysis is performed with R (version 4.0.1) [30] and In-Silico Online, version 2.3.0 [31]. Difference in means was assessed by Welch’s two-sided t-test, difference in median by Wilcoxon’s signed rank test with continuity correction, whereas the association between categorical variables was tested using the multinomial exact test [32]. Survival analyses were performed using the logrank test and multivariable analysis by Cox proportional hazards function, with death by cancer as an endpoint. An alpha level of 0.05 was applied to determine statistical significance. Arithmetic mean values are presented together with standard deviations (s).”

  1. The authors must take care of proper abbreviations and explanations. E.g. on line 161 what is denoted by s (s=7%, s=6.1%)?

In response to the reviewer's comment, we have explained the following abbreviation in the Materials and Methods of the manuscript on page 4 as follows:

"Arithmetic mean values are presented together with standard deviations (s). "

  1. Figure 1: Why are auto-fluorescence images are shown for K, ER, and CD45 when the cells were actually labelled with fluorescent dyes? In addition, the background correction across the images seems to be different. Is this due to permeabilization? The authors may provide more technical description / validation regarding the improved detectability.

Breast cancer cells have to be negative for the common leukocyte marker CD45 as evidenced by negative CD45 images. Moreover, we show that the intensity of ER immunofluorescence can differ among the MCF7 cells from negative to strong. CellSearch uses an automatic background correction leading to a dark background when the antigen is strongly expressed and a gray noise one in case of absent or low expression. Distinguishing a low intensity of antigen-specific immunofluorescence from autofluorescence is difficult and dependent on the cell line cells analyzed.

In order to respond to the reviewer's comment, we adjusted the legend of Figure 1 as follows:

"Figure 1 - ER expression detected in breast cancer cells by the CellSearch System. SK-BR-3 (top row) is an ER-negative cell line and was used as a negative control. MCF-7 is ER-positive and examples are given for strongly positive (second row), weakly positive (third row), and negative (bottom row) nuclear staining. Standard selection markers of the CellSearch System for tumor cells are keratin (K) and DAPI, whereas CD45 is used as an exclusion marker. Automatic low signal compensation leads to high background signal seen in CD45 and ER-negative scans. Pictures were taken using 10X magnification.”

  1. Table 1: G-test is not appropriate choice for such small dataset. The test is recommended for large datasets (>1000 values) with independent samples. Please use chi-square test.

We recognize the concern of the reviewer, that the G-test may be inaccurate with a small sample size, even with the Williams correction. However, both the Chi-square test and G-test provide estimations only (a common misconception is that the Chi-square test is still an appropriate statistical test for contingency tables. The test, first introduced in 1900 by Pearson, was designed to make a computationally easy inferences in a time in which computers were not available). Therefore, we recalculated the statistical significances using the multinomial exact tests. We have altered the p-values throughout the manuscript and adjusted the Materials and Methods, section Statistics, on page 3 as follows:

Difference in means was assessed by Welch’s two-sided t-test, difference in median by Wilcoxon’s signed rank test with continuity correction, whereas the association between categorical variables was tested using the multinomial exact test [32].

Because the conclusions did not change, no further adjustments to the manuscript were necessary.

  1. Table 1: Please fix typos (e.g. 3-Jan, 4-Mar etc…)

We thank the reviewer for spotting this oversight, Table 1 on page 5 has been corrected.

  1. Figure legends must have more details e.g. magnification, number of values, patients, what type of sample (baseline or under therapy) etc…

In order to respond to the reviewer's comment, we adjusted the legends as follows:

"Figure 1 - ER expression detected in breast cancer cells by the CellSearch System. SK-BR-3 (top row) is an ER-negative cell line and was used as a negative control. MCF-7 is ER-positive and examples are given for strongly positive (second row), weakly positive (third row), and negative (bottom row) nuclear staining. Standard selection markers of the CellSearch System for tumor cells are keratin (K) and DAPI, whereas CD45 is used as an exclusion marker. Automatic low signal compensation leads to high background signal seen in CD45 and ER-negative scans. Pictures were taken using 10X magnification."

" Figure 2 - Number of ER-positive and -negative CTCs. Patient blood samples in which CTCs were detected (n=31). Depicted are the number of ER-positive (black) and ER-negative (gray) CTCs per sample.”

" Figure 3 - Change in CTC number. Boxplots showing the distribution of the change in the number of CTCs at progression of disease (n=14) and at stable disease (n=11)."

  1. Figure 2. is missing!

We thank the reviewer for spotting this error, Figure 2 was added in the main text of the manuscript on page 6.

Reviewer 4 Report

Ningsi and colleagues performed an interesting study to investigate clinical significance of ER+ CTCs in metastatic breast cancer patients. Some key points need to be addressed by authors:

1. Because heterogeneous expression of ER in the primary tumor and/or CTCs might be dynamic, did authors try to continuously detect ER expression on CTCs in patients?

2. In this study, the CellSearch strategy was restricted to detect only EpCAM+/CK+/ER+ CTCs. However, a significant amount of CTCs may be EpCAM negative due to EMT, and some of those CTCs may express ER. Authors should address this critical point in Discussion and propose a potential alternative strategy to readers, based upon published methods (Li et al. 2018 Cancer Res 24:5261), to compensate the drawback or limitation of the CellSearch technique.

Author Response

Point-by-point reply to the comments raised by the reviewers with respect to the manuscript " Detection and characterization of estrogen receptor α expression of circulating tumor cells as a prognostic marker" by Ningsi et al.

We would like to thank the reviewers for their time and effort to provide their reviews. We have carefully revised the manuscript according to the suggestions provided.

All modifications in the revised manuscript are marked up using the "Track Changes" function.

Reviewer 4 - Comments for the Author.

Ningsi and colleagues performed an interesting study to investigate clinical significance of ER+ CTCs in metastatic breast cancer patients. Some key points need to be addressed by authors:

  1. Because heterogeneous expression of ER in the primary tumor and/or CTCs might be dynamic, did authors try to continuously detect ER expression on CTCs in patients?

The reviewer addresses an important point here and we acknowledge the dynamic expression of ER in cancer. So far, all methods for CTC detection and characterization are generally based on extracting the CTCs from the blood, fixing them, and finally staining the cells for determining the expression of certain markers. Hence, only a snapshot of the status of the marker under investigation is possible with currently technologies. Only recently, we have been able to establish the first cell line derived from CTCs from ER-positive breast cancer [Koch et al. 2020], making it possible to study ER expression of the same cells over time. In our article, we describe a stable ER expression of this cell line, as well as its dependency on ER for growth and proliferation. Although this is a single case only, it does indicate that ER expression may also be stable among CTCs. Further work is warranted with larger cohorts of breast cancer patients, studying the ER expression in CTCs closely over time. In order to address the comments raised by the reviewer, we state in the Discussion on page 10 the following:

Recently, we established a permanent CTC line (CTC-ITB-01) derived from ER-positive breast cancer patient. Performing this cell line will facilitate the study of ER expression on a large scale providing further insights into the dynamics of ER expression in therapy response [35]. Overall, the identification and monitoring of ER status of CTCs is extremely important for the management of breast cancer patients. Further investigation towards a robust and reproducible assay is needed with larger cohorts.”

  1. In this study, the CellSearch strategy was restricted to detect only EpCAM+/CK+/ER+ CTCs. However, a significant amount of CTCs may be EpCAM negative due to EMT, and some of those CTCs may express ER. Authors should address this critical point in Discussion and propose a potential alternative strategy to readers, based upon published methods (Li et al. 2018 Cancer Res 24:5261), to compensate the drawback or limitation of the CellSearch technique.

In response to the reviewer’s comment, we addressed the following point in the Discussion of the manuscript on page 10 as follows:

As a consequence of the limitations of the study, the relatively low number of patients with detectable ER-positive CTCs, the relatively short follow-up time, and heterogeneity in sample collection time-points in our study, an association with survival or clinical variables such as tumor burden and grade could not be made with ER status of CTCs yet. Furthermore, downregulation of EpCAM in CTCs that underwent epithelial-mesenchymal transition (EMT) may result in missing critical tumor cell subpopulations. Previously, we established an alternative approach based on EpCAM-free detection to overcome this limitation and capture as many CTCs as possible [24]. Nevertheless, to our knowledge, this is one of few studies that evaluating ER status of CTCs using the CellSearch CXC kit. An important finding of this study is the requirement of permeabilization in order for ER antibodies to penetrate the cell membrane and reach the cell nucleus. Therefore, future studies conducted using the CellSearch System should make use of the CellSave preservation tubes for blood collection.

Round 2

Reviewer 2 Report

Authors have addressed all concerns.

Reviewer 3 Report

I have no further comments